# Formation of LaNi$_5$ Hydrogen Storage Alloy by Electrodeposition of La Using Molten Salt

**Michihisa Fukumoto * , Kano Nakajima and Hiroki Takahashi**

Department of Materials Science, Graduate School of Engineering Science, Akita University, Akita 010-8502, Japan
* Correspondence: fukumoto@gipc.akita-u.ac.jp

**Abstract:** A hydrogen storage alloy was formed by electrodepositing La using a molten salt. La was electrodeposited using Ni as a substrate in NaCl-KCl-5.0 mol% LaF$_3$ molten salt at electrodeposition temperatures of 750 °C and 900 °C. The electrodeposition potential was $-2.25$ V. The LaNi$_5$ hydrogen storage alloy was then prepared by the electrodeposition of La and the mutual diffusion of the Ni substrate. As a result, it was clarified that La can be electrodeposited by using a molten salt. Single-phase LaNi$_5$ was produced at 750 °C rather than at 900 °C. It became possible to uniformly form LaNi$_5$, an intermetallic compound, on the substrate surface. The prepared hydrogen storage alloy was exposed to Ar-10%H$_2$ to store hydrogen; at this time, hydrogen was stored by changing the sample temperature. The discharged hydrogen was measured by a gas sensor. It was clarified that the hydrogen storage and hydrogen discharge were the highest in the sample obtained by electrodepositing La for 1 h at 750 °C. LaNi5 formed by electrodeposition showed hydrogen storage properties, and this method was found to be effective even for samples with complex shapes.

**Keywords:** molten salt; LaNi$_5$; hydrogen; gas sensor; electrodeposition

---

## 1. Introduction

Hydrogen is attracting attention as an alternative energy source to fossil fuels [1]. It is necessary to liquefy hydrogen in order to transport a large amount of it; however, it must be cooled to $-253$ °C in order to be liquefied. Therefore, different transportation methods must be established. It is conceivable that hydrogen can be stored in a hydrogen storage alloy and transported. Many studies of hydrogen storage alloys have already been reported. Mechanical alloying methods [2–8], powder synthesis [9–12] and mechanical grinding [13,14] are available as methods for producing hydrogen storage alloys. However, they need to be effectively used since La is a rare element. Furthermore, LaNi$_5$ is an intermetallic compound and its composition range is narrow, so it is difficult to prepare it. Another problem is that it is difficult to process. Therefore, the authors of this paper decided to solve these problems by producing a LaNi$_5$ hydrogen storage alloy by an electrodeposition method. This method can produce LaNi$_5$ with a large surface area. There have been reports on surface modification [15,16] and the effects of tertiary elements [17–19]. The method proposed here may replace previously proposed methods.

The authors have easily electrodeposited metals using molten salts, which are high-temperature liquids. An alloy of an electrodeposited metal and a substrate can be formed on the surface since electrodeposition is performed at higher temperatures. Therefore, the authors considered making a hydrogen storage alloy using this method. Figure 1 shows the method for producing the hydrogen storage alloys used in this study. First, La is electrodeposited on the substrate metal using a molten salt. At this time, the electrodeposited La reacts with the substrate metal to form an alloy in order to carry out the experiment at high temperature. The authors have clarified that various metals are electrodeposited on the Ni substrate to form an alloy surface layer by using high-temperature molten salt. Therefore, the LaNi$_5$ hydrogen storage alloy was produced by electrodepositing La using

---

a molten salt to form an alloy of Ni and La. Molten salt electrodeposition is less affected by the shape of the sample than the dry process [20–22], so it is suitable for processing complicated shapes.

Creation method by molten salt electrodeposition

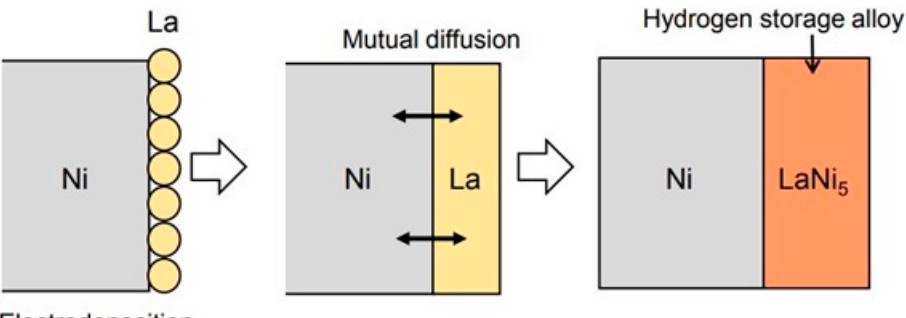

**Figure 1.** Schematic diagram of formation of LaNi$_5$ hydrogen storage alloy by La electrodeposition.

In addition, the hydrogen storage characteristics of the hydrogen storage alloy produced by electrodeposition were measured by a gas sensor. There are few reports about the effect of temperature on the absorption of hydrogen [23–25]. The reason is that it is difficult to measure the hydrogen storage capacity in situ. The authors measured hydrogen in situ using yttria-stabilized zirconia, a solid electrolyte. As a result, it was clarified that even a small amount of hydrogen can be accurately measured in situ.

Therefore, the purpose of this study was to generate the LaNi$_5$ hydrogen storage alloy by La electrodeposition in a molten salt and to evaluate the hydrogen storage characteristics by in situ measurement using a gas sensor. In addition, the hydrogen storage capacity was increased by changing the shape of the sample and increasing the surface area.

## 2. Experimental Procedure

### 2.1. La Electrodeposition Using the Molten Salt

Commercially available 99.9% pure Ni (The Nilaco Corporation, Tokyo, Japan) was used as the substrate. A plate-shaped sample was used as the sample morphology, and a mesh-shaped sample was used to increase the surface area. The surface of the plate-shaped sample was polished to No. 800 with emery paper, then cleaned ultrasonically in acetone. The mesh-shaped sample was washed with acetone without polishing and was used in the experiment.

The LaNi$_5$ hydrogen storage alloy was formed by electrodepositing the La. The electrodeposition was carried out using a molten salt. The NaCl-KCl mixed salt, having an equimolar composition to which 5.0 mol% LaF$_3$ was added, was used as the electrolytic bath. The electrolytic cell used in this experiment was described in a previous report [26]. The mixed salt of NaCl-KCl-AgCl (45:45:10 mol%) was placed in a mullite tube, and an Ag wire was immersed in the mixed salt as the reference electrode. The bath temperatures during the La electrodeposition were 750 °C and 900 °C. Ar gas was added to the cell at the flow rate of 200 mL min$^{-1}$ during the experiment. The electrodeposition was −2.25 V, at which the reduction reaction of La occurs based on the cathode polarization curve. After the treatment, the sample was removed from the bath and the salt, which had adhered to the sample surface, was removed by washing with water. The cross-section of the sample after treatment was observed and analyzed using a scanning electron microscope (scanning electron microscope: SEM, Tokyo, Japan) and an X-ray micro-analyzer (electron probe micro-analyzer: EPMA, Tokyo, Japan). Furthermore, the identification of the electrodeposited layer was performed by the X-ray diffraction method. Cu$K\alpha$ radiation was used as the X-ray source.

### 2.2. Measurement of Hydrogen Storage Capacity by Solid Electrolyte

The LaNi$_5$ hydrogen storage capacity produced by electrodeposition was observed in situ using a gas sensor. Figure 2 shows a schematic diagram of the equipment used to measure the hydrogen storage capacity. A sample obtained by electrodepositing La on the surface was placed in an electric furnace. The sample temperature was set by moving the electric furnace up and down. The sample was heated by flowing Ar-10% H$_2$ gas at 40 mL min$^{-1}$ to occlude the hydrogen. The temperatures at which the hydrogen was occluded were 340 °C, 440 °C and 540 °C. Next, Ar gas was flowed and the sample was heated at 340 °C to discharge the hydrogen. The amount of hydrogen discharged was measured using a gas sensor called an oxygen pump sensor.

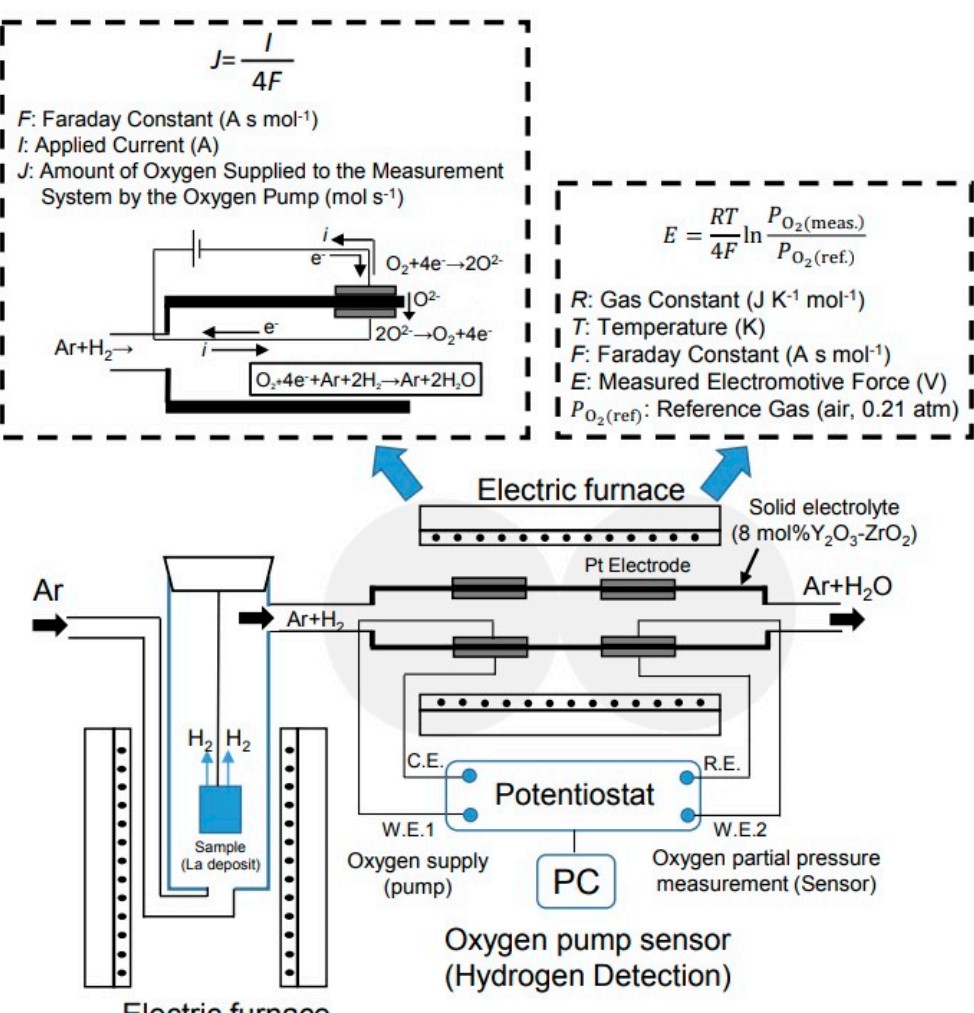

**Figure 2.** Diagram of measuring device for hydrogen storage capacity of LaNi$_5$ hydrogen storage alloy.

Tubular yttria-stabilized zirconia (8 mol% Y$_2$O$_3$-ZrO$_2$) was used for the oxygen pump sensor. The measurement gas was then flowed inside the tube. The oxygen pump sensor consisted of a sensor part and a pump part. Pt was used as the electrode. The oxygen partial pressure can be measured by this sensor part. The oxygen partial pressure was determined using the Nernst equation. The electromotive force measured by the oxygen sensor was substituted into the Nernst equation shown in Equation (1), and the oxygen partial pressure was obtained.

$$E = \frac{RT}{4F} \ln \frac{P_{O_2(\text{meas.})}}{P_{O_2(\text{ref.})}} \tag{1}$$

where $R$ indicates the gas constant (J K$^{-1}$ mol$^{-1}$); $T$ indicates the temperature (K); $F$ indicates the Faraday constant (A s mol$^{-1}$); $E$ indicates the measured electromotive force (V); and $P_{O_2(ref)}$ indicates the reference gas (air, 0.21 atm). The sensor was operated at 850 °C. The oxygen can be supplied to the inside of the pipe by the reaction of (2) by passing an electric current through the pump part, because this gas sensor is an oxide ion conductor.

$$2O^{2-} \rightarrow O_2 + 4e^- \tag{2}$$

The hydrogen discharged from the hydrogen storage alloy flows together with Ar, which is a carrier gas. The oxygen in Equation (2) and the released hydrogen react as in Equation (3) by applying an electric current in the pump section.

$$O_2 + 4e^- + 2H_2 \rightarrow 2H_2O \tag{3}$$

The sensor unit maintains the initial oxygen partial pressure state. When hydrogen is released from the hydrogen storage alloy, the oxygen partial pressure in the pipe decreases. Oxygen is supplied from the outside of the pipe by the pump part to maintain the initial state. As a result, the amount of hydrogen generated can be determined from the amount of supplied oxygen. More emitted hydrogen indicates more supplied oxygen. The amount of hydrogen discharged from the hydrogen storage alloy can be found by measuring the current value of the oxygen supply. The supplied oxygen was calculated using Faraday's law [27].

### 3. Results and Discussion

#### 3.1. Cathode Polarization Curve in Isomolar NaCl-KCl Composition with Added LaF$_3$

Figure 3 shows the cathode polarization curve at 750 °C in the NaCl-KCl-5.0 mol% LaF$_3$ molten salt on a Ni substrate. The results of the cathode polarization curve in the NaCl-KCl molten salt without the addition of LaF$_3$ are also shown for comparison. The cathode current increased from around −1.8 V with no addition. This is considered to be an increase in the cathode current due to the reduction reaction of Na$^+$ and K$^+$ contained in the molten salt. On the other hand, an increase in the cathode current was also observed from around −1.8 V even in the bath to which LaF$_3$ was added. However, the cathode current increased more than the result without adding LaF$_3$ from around −2.25 V. It is considered that the reduction reaction of La$^{3+}$ occurred from around −2.25 V. Therefore, La electrodeposition was performed at −2.25 V.

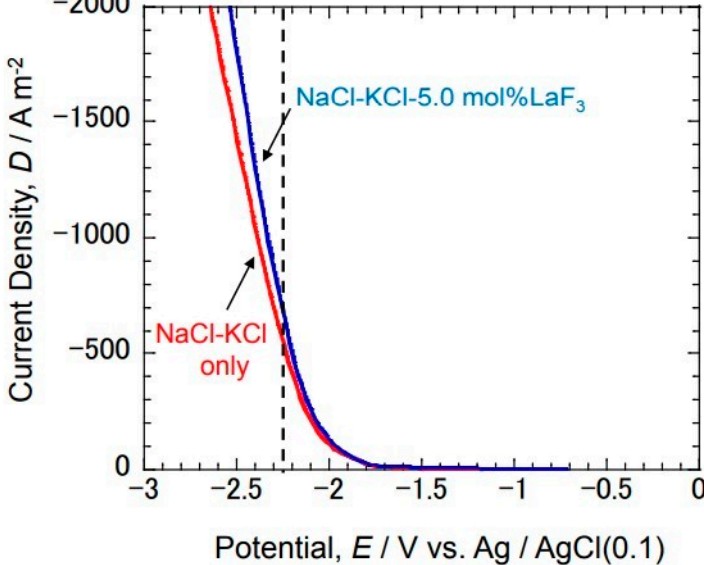

**Figure 3.** Cathode polarization curves of Ni in NaCl-KCl-5.0 mol%LaF$_3$ melts at 750 °C.

### 3.2. Cross-Sectional Microstructure of the Sample Obtained by Electrodepositing La

Figure 4 shows the cross-sectional microstructure and elemental analysis results after the La electrodeposition at −2.25 V for 1 h and 1.5 h in the molten salt at 750 °C. An electrodeposited layer of about 7.8 μm was observed on the surface of the 1 h sample. As a result of analyzing this layer, La was 16.9 at.% and Ni was 83.1 at.%. Here, the LaNi$_5$ hydrogen storage alloy was formed. On the other hand, an electrodeposited layer of about 5.6 μm was observed on the surface of the sample with an electrodeposition time of 1.5 h. As a result of analyzing this electrodeposition layer, La was 17.6 at.% and Ni was 82.4 at.%. It is considered that LaNi$_5$, which has a composition similar to that of a hydrogen storage alloy, can be produced in a sample with an electrodeposition time of 1 h. The longer the electrodeposition time, the higher the La concentration. The difference in thickness was considered to be within the margin of error.

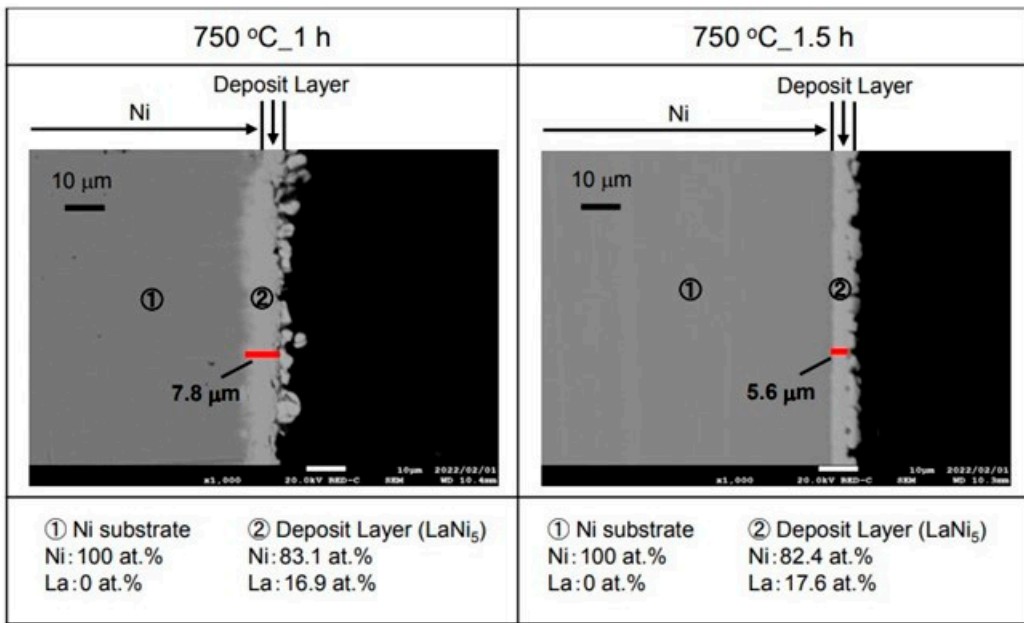

**Figure 4.** Cross-sectional microstructure and element analysis of Ni after La deposition at −2.25 V in NaCl-KCl-5.0 mol%LaF$_3$ melts at 750 °C.

Figure 5 shows the cross-sectional microstructure and element analysis results after the La electrodepositions at −2.25 V for 1 h and 1.5 h in the molten salt at 900 °C. The electrodeposition layer of about 5.8 μm was observed in the inner layer for the 1 h sample. When this layer was analyzed, La was 17.3 at.% and Ni was 82.7 at.%. It is considered that LaNi$_5$, which is a hydrogen storage alloy, was formed. However, La was electrodeposited on the outer layer. For the sample with the electrodeposition time of 1.5 h, an electrodeposited layer thicker than the 1 h one was observed on the surface. This electrodeposition layer also had a two-layer structure. As a result of the analysis of the inner layer, La was 16.9 at.% and Ni was 83.1 at.%. It appears that the LaNi$_5$ hydrogen storage alloy was formed in this inner layer like the other samples. However, La was 99.5 at.% and Ni was 0.5 at.% in the outer layer according to our analysis. Therefore, it appeared to be a La metal. Furthermore, many cracks were observed in the La metal for both the 1 h and 1.5 h samples.

Figure 6 shows the XRD diffraction results of the La-deposited samples (1 h and 1.5 h) at 750 °C and 900 °C. The peaks of LaNi$_5$ and Ni were observed under the electrodeposition conditions of 1 h at 750 °C. It is probable that a peak was observed on the substrate due to the thin electrodeposition layer. In addition, the generated electrodeposition layer could be identified as LaNi$_5$, which is a hydrogen storage alloy. Similar results were obtained for the 1.5 h sample at 750 °C. However, the peak of La was observed at 900 °C. The La metal and the hydrogen storage alloy LaNi$_5$ were electrodeposited for 1 h at 900 °C. This result

is consistent with the result of the cross-section observation. In addition, it is considered that the peaks of LaNi$_5$ and Ni were observed in the sample at 900 °C for 1.5 h because there was a part without any La metal. The reaction slowly proceeds at 750 °C, resulting in a uniform electrodeposition layer, but the reaction is rapid at 900 °C, resulting in a non-uniform electrodeposition layer.

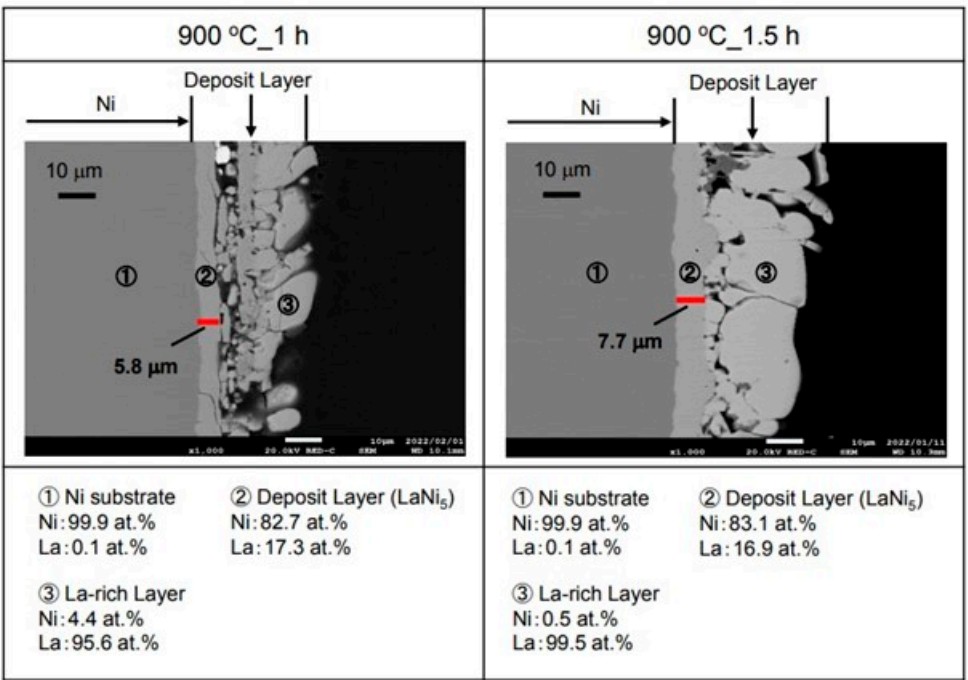

**Figure 5.** Cross-sectional microstructure and element analysis of Ni after La deposition at −2.25 V in NaCl-KCl-5.0 mol%LaF$_3$ melts at 900 °C.

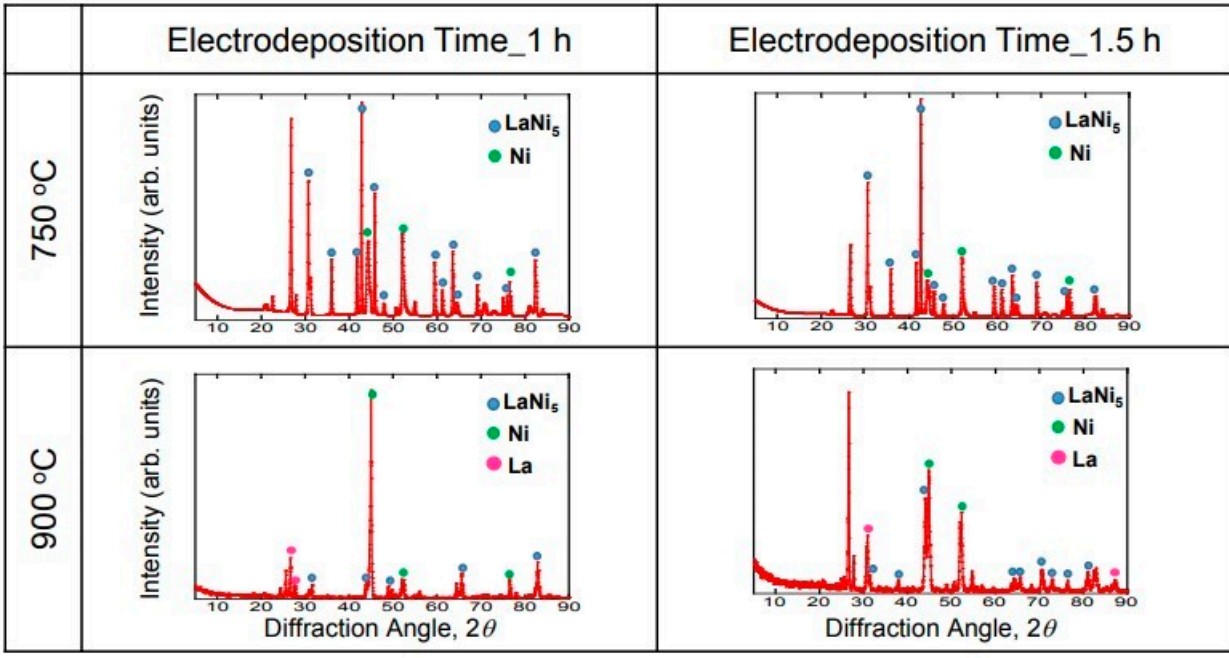

**Figure 6.** X-ray diffraction patterns of Ni after La deposition at −2.25 V in NaCl-KCl-5.0 mol%LaF$_3$ melts at 750 °C and 900 °C.

### 3.3. Measurement of Occluded and Discharged Hydrogen by Gas Sensor

Figure 7 shows the results of the temperature dependence of the oxygen partial pressure (a) and the time dependence of the amount of discharged hydrogen (b) measured by the gas sensor. Figure 7a shows the result of measuring the oxygen partial pressure by changing the temperature after occluding hydrogen at 540 °C in Ar-10% $H_2$. The decrease in the oxygen partial pressure indicates that hydrogen is being generated. Therefore, it was clarified that hydrogen was not occluded in the 900 °C sample because the oxygen partial pressure did not decrease. On the other hand, a decrease in the oxygen partial pressure was observed from around 200 °C in the sample deposited at 750 °C for 1 h. The oxygen partial pressure decreased the most at 250 °C. On the other hand, a decrease in the oxygen partial pressure was observed from around 250 °C, and the oxygen partial pressure decreased the most at 280 °C in the sample electrodeposited at 750 °C for 1.5 h. It can be seen that the hydrogen emission was completed at 340 °C in both samples. The hydrogen discharge was measured at 340 °C, at which the hydrogen discharge was completed in all the samples. Figure 7b shows the result of measuring the amount of discharged hydrogen by a gas sensor when the hydrogen was discharged at 340 °C. It can be seen that hydrogen is rapidly discharged when the sample is heated to 340 °C. However, the hydrogen discharge was low using the 900 °C sample. On the other hand, a large peak is shown at 750 °C, and hydrogen was discharged for a long time, especially in the sample electrodeposited at 750 °C for 1 h. Therefore, it was found that most of the hydrogen was occluded.

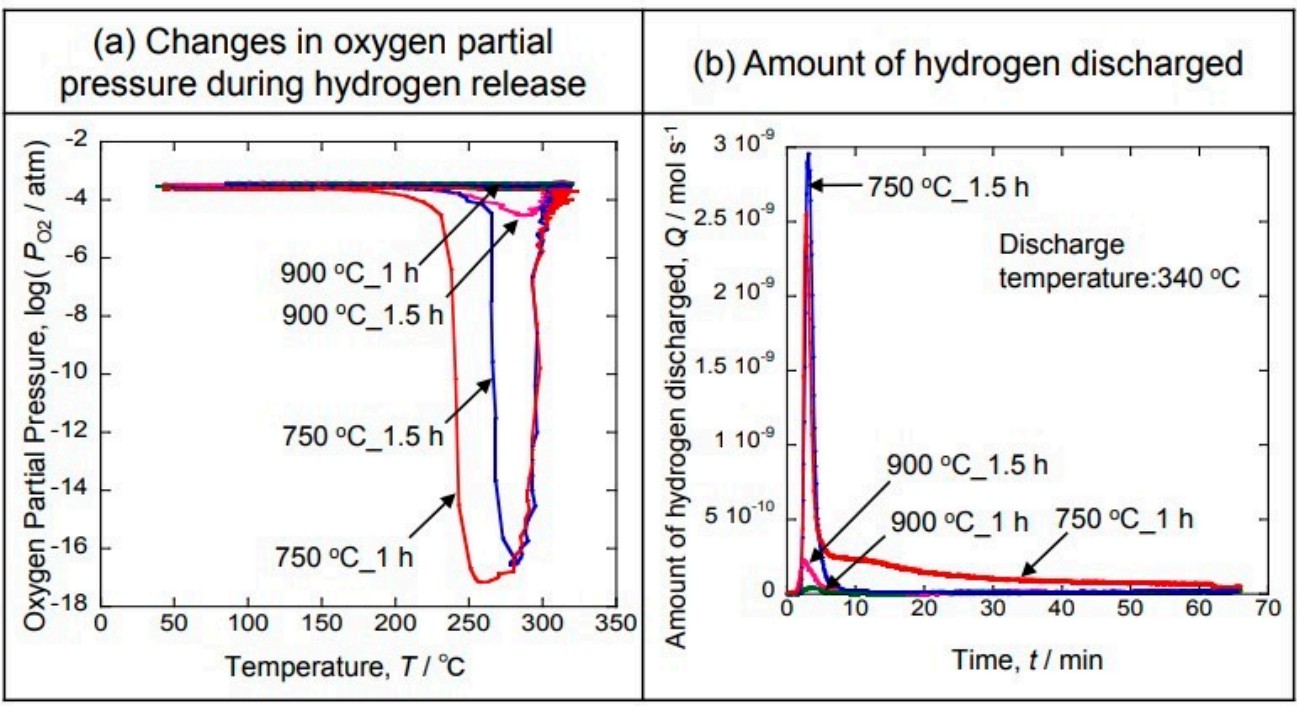

**Figure 7.** Oxygen partial pressure and amount of charged hydrogen of Ni after La deposition at −2.25 V in NaCl-KCl-5.0 mol%LaF$_3$ melts at 750 °C and 900 °C.

Figure 8 shows the time-dependent results of the oxygen partial pressure when hydrogen was discharged by changing the temperature of the hydrogen storage experiment to 340 °C, 440 °C and 540 °C. The hydrogen discharge experiment was performed at 340 °C. A decrease in the oxygen partial pressure was observed with the generation of hydrogen for the 750 °C sample. However, no decrease in the oxygen partial pressure was observed with hydrogen generation for the 900 °C sample. On the other hand, a decrease in the oxygen partial pressure due to the generation of hydrogen was observed in the samples in

which hydrogen was occluded at 440 °C and 540 °C for the 750 °C samples. In addition, the 440 °C and 540 °C results displayed similar behavior. Therefore, it was clarified that hydrogen was occluded at 440 °C. When hydrogen was discharged at 340 °C, a decrease in the oxygen partial pressure associated with hydrogen generation was observed at 1.5 h, but not at 1 h. Therefore, 340 °C appears to be the critical temperature.

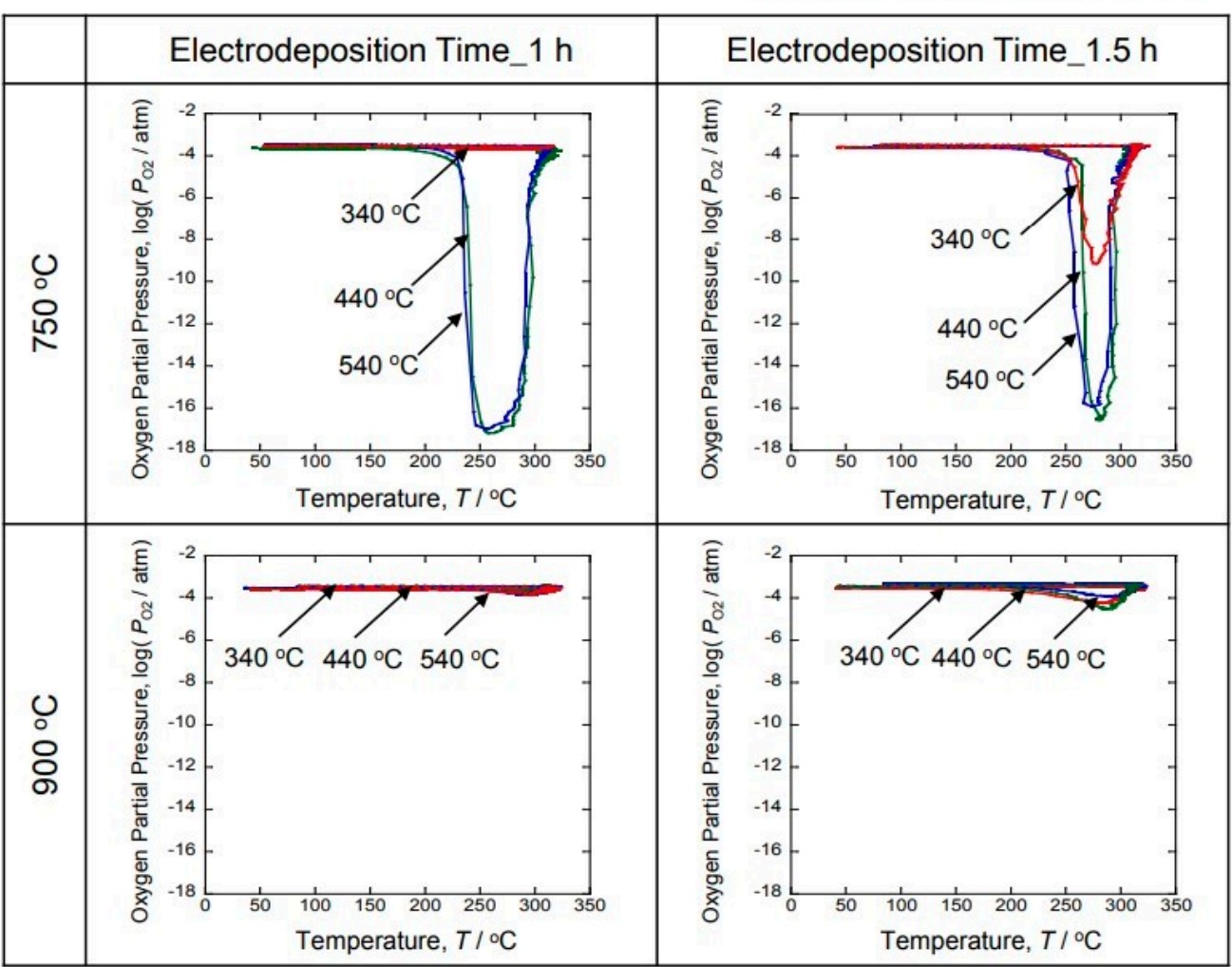

**Figure 8.** Oxygen partial pressure–temperature curves of Ni after La deposition at −2.25 V in NaCl-KCl-5.0 mol%LaF$_3$ melts at 750 °C and 900 °C. (Hydrogen storage temperature: 340 °C, 440 °C and 540 °C.)

Figure 9 shows the hydrogen discharge measured by the gas sensor. The hydrogen discharge experiment was performed at 340 °C. In addition, the temperature at which hydrogen was occluded was changed to 340 °C, 440 °C and 540 °C. No hydrogen emission was observed for the sample with the electrodeposition temperature of 900 °C. Hydrogen emission was observed for the sample with the electrodeposition temperature of 750 °C. The large peak due to hydrogen generation can be observed during the initial stage for the sample with the electrodeposition time of 1 h. The hydrogen storage temperature peaked at 440 °C and 540 °C. After showing a large peak, the amount of hydrogen generated sharply decreased and remained low. On the other hand, a sample with the electrodeposition time of 1.5 h also showed a large amount of hydrogen generation during the initial stage. In particular, the peak increased at 440 °C and 540 °C. Therefore, 440 °C and 540 °C resulted in a large amount of hydrogen storage. After the discharge, the hydrogen storage amount

was low. Based on these results, it was clarified that a large amount of hydrogen was emitted during the initial stage.

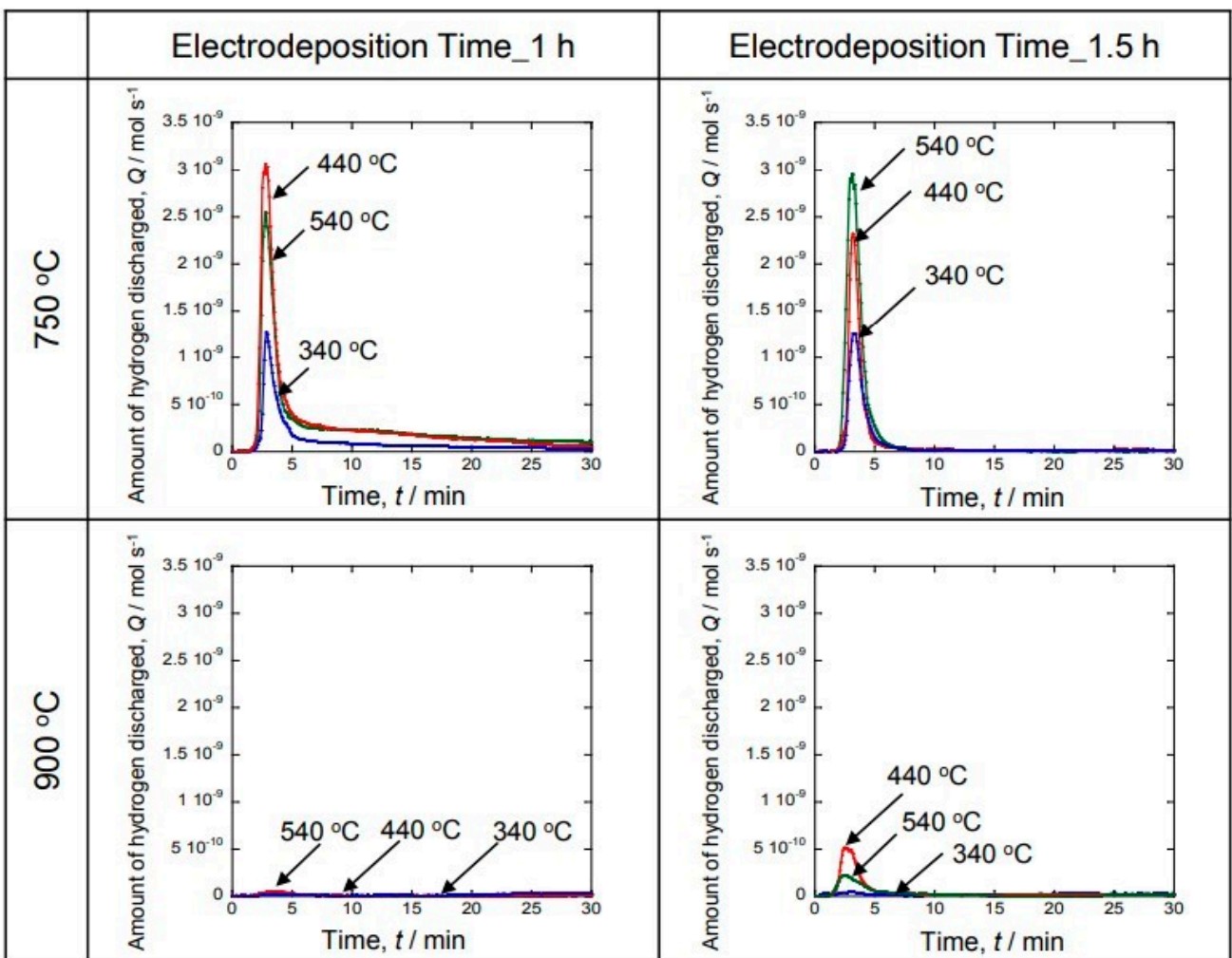

**Figure 9.** Hydrogen discharge–temperature curves of Ni after La deposition at −2.25 V in NaCl-KCl-5.0 mol%LaF$_3$ melts at 750 °C and 900 °C. (Hydrogen storage temperature: 340 °C, 440 °C and 540 °C.)

Figure 10 shows the Arrhenius plot of the total amount of hydrogen generated, calculated from the results of Figure 9. A high value was shown for the sample in which the electrodeposition layer was formed at 750 °C. On the other hand, the value was low in the sample for which the electrodeposition layer was prepared at 900 °C. Moreover, it was clarified that hydrogen was not occluded at 900 °C, since no temperature dependence was observed. On the other hand, the 1 h sample at 750 °C showed a temperature dependence and the slope became smaller. Therefore, it can be easily seen that hydrogen storage and discharge occur. The 1.5 h sample showed a small inclination, but the inclination was greater than that of the 1 h sample. It is considered that a more uniform hydrogen storage alloy was produced from the 1 h sample.

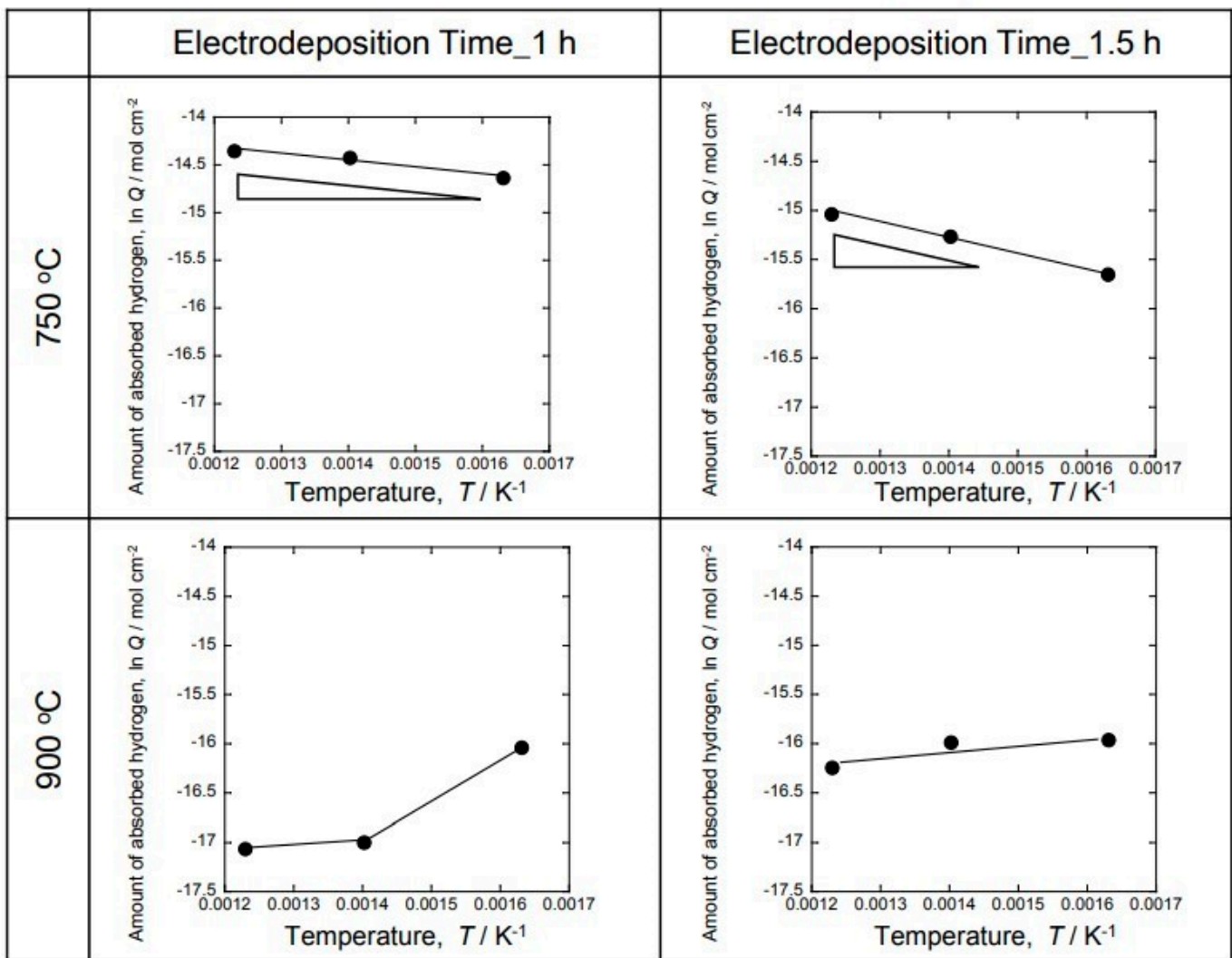

**Figure 10.** Arrhenius plots of Ni after La deposition at −2.25 V in NaCl-KCl-5.0 mol%LaF$_3$ melts at 750 °C and 900 °C.

*3.4. Hydrogen Storage Alloy Using Ni Mesh*

Figure 11 shows the results of the La electrodeposition using a Ni mesh to increase the surface area and increase the hydrogen storage capacity. Figure 11a shows the cross-sectional microstructure using 20 mesh Ni. In addition, (b) shows the cross-sectional microstructure using 100 mesh Ni. The electrodeposition layer was 3.6 μm in the 20-mesh sample of (a). As a result of analyzing this electrodeposition layer, La was 16.5 at.% and Ni was 83.5 at.%. It is considered that LaNi$_5$, which is a hydrogen storage alloy, was produced. The 100-mesh sample for (b) was 5.8 μm-thick, which is thicker than that of the 20-mesh sample. The analysis of this electrodeposition layer revealed that La was 17.0 at.% and Ni was 83.0 at.%. It is considered that LaNi$_5$, which is a hydrogen storage alloy, was produced in the same way as the 20-mesh sample.

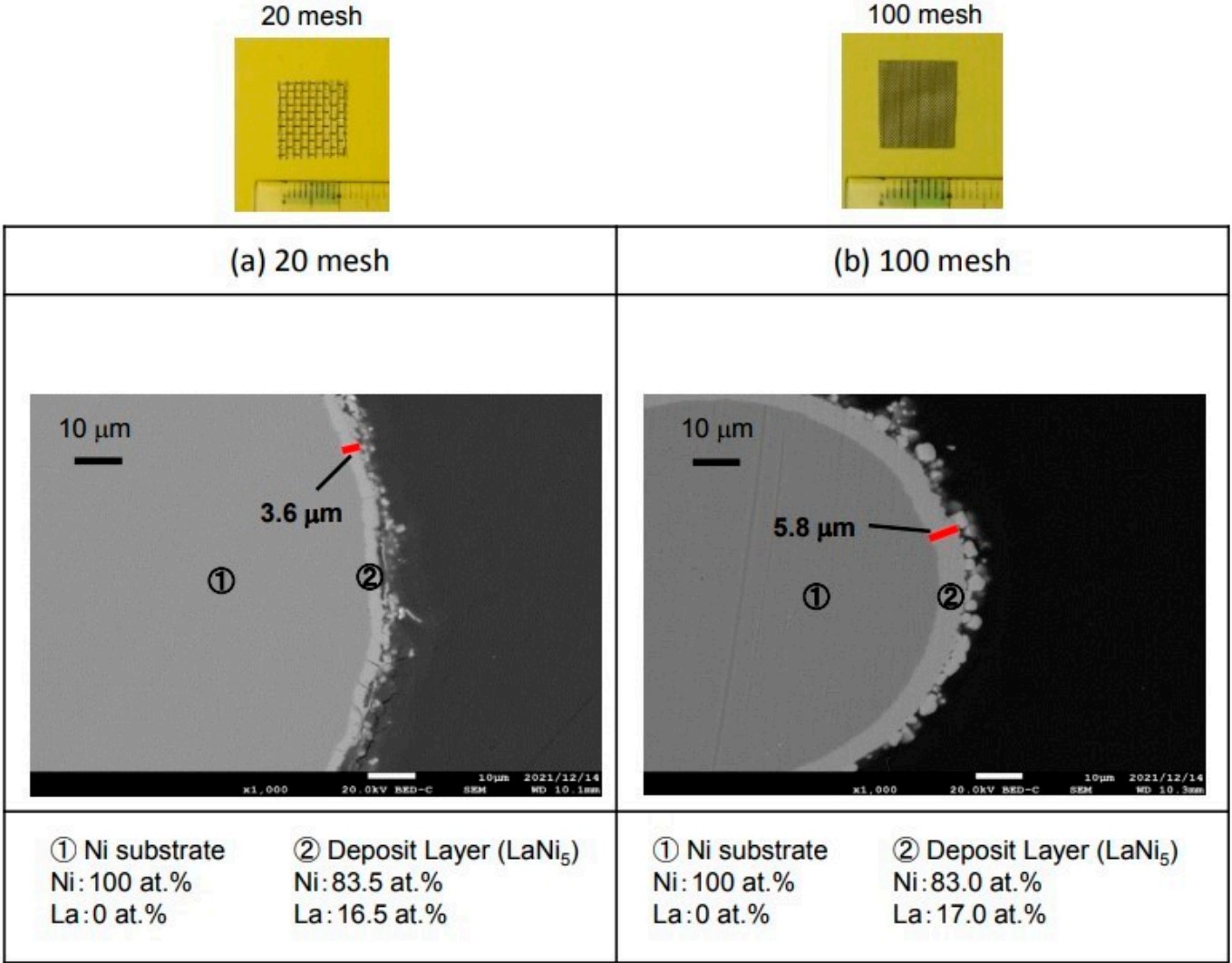

**Figure 11.** Cross-sectional microstructure and element analysis of Ni mesh after La deposition at −2.25 V in NaCl-KCl-5.0 mol%LaF$_3$ melts at 750 °C.

Figure 12 shows the amount of hydrogen generated measured by a gas sensor when hydrogen was occluded at 540 °C and discharged at 340 °C. The table shows the total amount of hydrogen obtained by integrating Figure 12a. It was found that the performance of the mesh-shaped sample was improved compared to that of the plate-shaped sample. In particular, the amount of generated hydrogen was dramatically improved for the 100-mesh sample. The initial amount of hydrogen generated dramatically increased for the 100-mesh sample. It is considered that the surface area increased and the hydrogen storage capacity increased when the mesh-shaped sample was used. Furthermore, it can be seen that the mesh-shaped sample occludes more hydrogen than the plate-shaped sample when comparing the amount of hydrogen obtained by integrating Figure 12a. In this experiment, the hydrogen storage time was 1 h and Ar-10% H$_2$ gas was used. However, if pure hydrogen gas were used, it is considered that more hydrogen could be occluded than that shown by the result obtained in this experiment.

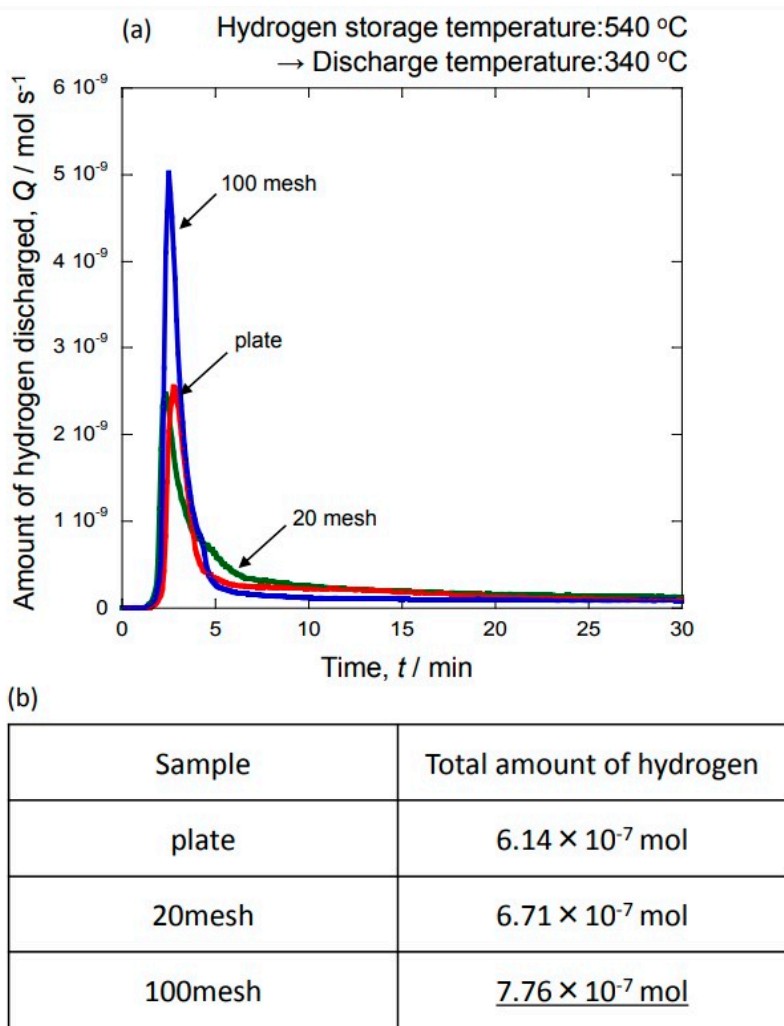

**Figure 12.** Amount of hydrogen discharged (**a**) and total amount of hydrogen (**b**) of Ni plate and Ni mesh after La deposition in NaCl-KCl-5.0 mol%LaF$_3$ melts at 750 °C.

## 4. Conclusions

The La electrodeposition was performed at −2.25 V in a NaCl-KCl-3.5 mol%LaF$_3$ molten salt using Ni as a substrate. The electrodeposition temperatures were 750 °C and 900 °C. The electrodeposition time was 1 h and 1.5 h. Furthermore, the stored and discharged hydrogen of the electrodeposited layer was measured in situ by a gas sensor.

1. The LaNi$_5$ hydrogen storage alloy can be produced by La electrodeposition on Ni in the molten salt.

2. At the electrodeposition temperature of 900 °C, the La metal was formed in the outer layer and the LaNi$_5$ hydrogen storage alloy was formed in the inner layer. On the other hand, when the electrodeposition temperature was 750 °C, a single LaNi$_5$ was produced.

3. As a result of measuring the oxygen partial pressure of the sample that occluded hydrogen using a gas sensor, it was observed that the oxygen partial pressure decreased with the hydrogen discharge. Specifically, the sample at 1 h at the electrodeposition temperature of 750 °C showed the lowest oxygen partial pressure.

4. As a result of measuring the amount of discharged hydrogen using an oxygen pump sensor, it was clarified that a large amount of hydrogen was discharged during the early stage of the temperature increase.

5. The surface area was increased and the amount of stored hydrogen was significantly improved by changing the morphology of the substrate Ni sample from a plate shape to a mesh shape.

6. In order to further improve the performance, it is necessary to try to grow thick LaNi₅ by changing the deposition conditions.

**Author Contributions:** Conceptualization, M.F., K.N. and H.T.; methodology, M.F.; validation, M.F., K.N. and H.T.; formal analysis, K.N.; investigation, K.N.; data curation, K.N.; writing—original draft preparation, M.F.; writing—review and editing, H.T.; visualization, M.F.; supervision, M.F.; project administration, M.F.; funding acquisition, M.F. All authors have read and agreed to the published version of the manuscript.

**Funding:** This work was primarily supported by the Iketani Science and Technology Foundation, ISTF, Japan.

**Institutional Review Board Statement:** Not applicable.

**Informed Consent Statement:** Not applicable.

**Data Availability Statement:** Data sharing is not applicable to this article.

**Conflicts of Interest:** The authors declare no conflict of interest.

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
