# Peer review of "Formation of LaNi5 Hydrogen Storage Alloy by Electrodeposition of La Using Molten Salt"

_coatings, doi:10.3390/coatings12091268_

Round 1

Reviewer 1 Report

The manuscript presents an interesting study about the electrodeposition of LaF3 molten salt on the surface of the nickel. The paper needs major revisions before it is processed further, some comments follow:

Abstract:

The abstract must be improved. The abstract must contain information about:

-        Background: Please highlight the novelty of the study;

-        Methods: Describe briefly the main methods used to obtain and characterize the coating.

-        Results and conclusions: Indicate the main conclusions or interpretations.

 Introduction

The introduction section must be improved. In the introduction section, a comprehensive and exhaustive review of the state of the art in the field of the study must be provided. Please introduce and discuss more previous works, and highlight the experiments and results published previously.

Also, multiple citations have been introduced in bulk form "[2-23]", "[2-4,7,9,15]", "[8,11,12,16]" , "[26-28]", "[30-33]", "[34-36]"   and not distributed in the text in accordance with the affirmations that must be supported. Please introduce citations in a specific position to ensure clear correspondence between the affirmations from the introduction section and the previous publication. Moreover, to avoid this type of citing, please cite review type of studies.

Experimental procedure

Please rename this section to Materials and Methods. Also, divide this section into two: one about materials and one about the methodology of deposition and the characterizing methods.

Conclusions

The conclusion section should be improved. Write research suggestions and limitations.

Author Response

I wrote the answer in a PDF file.

Reviewer 2 Report

Referee Report

on paper “ Formation of LaNi5 hydrogen storage alloy by electrodeposition of La using molten salt “ (coatings-1884161) by author Michihisa Fukumoto, Kano Nakajima, Hiroki Takahashi submitted to Coatings

This is interesting paper. It reports the preparation and investigation of the structure and electrochemical performance of the LaNi5 hydrogen storage alloy prepared by the electrodeposition of La and the mutual diffusion of the Ni substrate. La was electrodeposited using Ni as a substrate in the NaCl-KCl-5.0mol% LaF3 molten salt at electrodeposition temperatures of 750 ℃ and 900 ℃ with the electrodeposition potential of -2.25V. The prepared hydrogen storage alloy was exposed to Ar-10%H2 to store hydrogen, at which time, hydrogen was stored by changing the sample temperature. The discharged hydrogen was measured by a gas sensor. The hydrogen storage and hydrogen discharge were the highest in the sample obtained by electrodepositing La under the conditions of 1 h at 750 ℃. The presented experimental results are reliable without any doubts. However, I have some comments and additions. I would like to note a few points to improve the paper before it can be published:

1.   The authors should give in 1. Introduction the specific examples of the production of metal films:

(1). T. Zubar, V. Fedosyuk, D. Tishkevich, O. Kanafyev, K. Astapovich, A. Kozlovskiy, M. Zdorovets, D. Vinnik, S. Gudkova, E. Kaniukov, A.S.B. Sombra, D. Zhou, R.B. Jotania, C. Singh, S. Trukhanov, A. Trukhanov, The effect of heat treatment on the microstructure and mechanical properties of 2D nanostructured Au/NiFe system, Nanomaterials 10 (2020) 1077. https://doi.org/10.3390/nano10061077.

2.   The authors should provide in 1. Introduction the specific examples of the use of metal films:

(2). A.V. Trukhanov, S.S. Grabchikov, A.A. Solobai, D.I. Tishkevich, S.V. Trukhanov, E.L. Trukhanova, AC and DC-shielding properties for the Ni80Fe20/Cu film structures, J. Magn. Magn. Mater. 443 (2017) 142-148. https://doi.org/10.1016/j.jmmm.2017.07.053.

3.   The authors should mention in 1. Introduction some information about metallic composite materials are perspective for practical applications:

(3). S.A. Sharko, A.I. Serokurova, N.N. Novitskii, V.A. Ketsko, M.N. Smirnova, A.H. Almuqrin, M.I. Sayyed, S.V. Trukhanov, A.V. Trukhanov, A new approach to the formation of nanosized gold and beryllium films by ion-beam sputtering deposition, Nanomaterials 12 (2022) 470. https://doi.org/10.3390/nano12030470.

4.   The proposed 3 papers should be inserted in References.

The paper should be sent to me for the second analysis after the moderate revisions.

Author Response

I wrote the answer in a PDF file.

Round 2

Reviewer 1 Report

The manuscript can be published in the present form. 

Reviewer 2 Report

Referee Report

on paper “ Formation of LaNi5 hydrogen storage alloy by electrodeposition of La using molten salt “ (coatings-1884161-v2) by author Michihisa Fukumoto, Kano Nakajima, Hiroki Takahashi submitted to Coatings

This paper has been well corrected and it should be published.
